# GENERATIVE ADVERSARIAL NETWORKS FOR IMAGE STEGANOGRAPHY

**Denis Volkhonskiy[2,3], Boris Borisenko[3] and Evgeny Burnaev[1,2,3]**
[1]Skolkovo Institute of Science and Technology
[2]The Institute for Information Transmission Problems RAS (Kharkevich Institute)
[3]National Research University Higher School of Economics (HSE)
`dvolkhonskiy@gmail.com`, `bborisenko@hse.ru` and
`e.burnaev@skoltech.ru`

## ABSTRACT

Steganography is collection of methods to hide secret information ("payload") within non-secret information ("container"). Its counterpart, Steganalysis, is the practice of determining if a message contains a hidden payload, and recovering it if possible. Presence of hidden payloads is typically detected by a binary classifier. In the present study, we propose a new model for generating image-like containers based on Deep Convolutional Generative Adversarial Networks (DCGAN). This approach allows to generate more setganalysis-secure message embedding using standard steganography algorithms. Experiment results demonstrate that the new model successfully deceives the steganography analyzer, and for this reason, can be used in steganographic applications.

## 1 INTRODUCTION

Recently developed Generative Adversarial Networks (GAN, see Goodfellow et al. (2014)) are powerful generative models, the main idea of which is to train a generator and a discriminator network through playing a minimax game. In the image domain, for a dataset generated by some density $p_{data}(x)$ a generator $G$ attempts to approximate the image generating distribution and to synthesize as realistic image as possible, while a discriminator $D$ strives to distinguish real images from fake ones.

There are several modifications of GAN that can generate realistic images:

- Deep Convolutional Generative Adversarial Networks (DCGAN, see Radford et al. (2015)) — this model is a modification of a GAN, specialized for generation of images;
- Conditional GAN — it allows generating objects from a specified class, see Mirza & Osindero (2014);
- Generation of images from textual description, see Reed et al. (2016).

In the present study we apply the DCGAN model to the problem of secure steganography. We construct a special container-image generator, synthetic output of which is less susceptible to successful steganalysis compared to containers, directly derived from original images. In particular, we investigate whether this methodology allows to deceive a given steganography analyzer, represented by a binary classifier detecting presence of hidden messages in an image.

## 2 STEGANOGRAPHY

Steganography is the practice of concealing a secret message, e.g. a document, an image, or a video, within another non-secret message in the most inconspicuous manner possible. In this paper we consider a text-to-image embedding, with the text given by bit string. More formally, for a message $T$ and an image $I$, a steganography algorithm is a map $S : T \times I \to \hat{I}$, where $\hat{I}$ is an image, containing the message $T$, such that $\hat{I}$ can not be visually distinguished from $I$.

The most popular and easy-to-implement algorithm of embedding is the Least Significant Bit (LSB) algorithm. The main idea of LSB is to store the secret message in the least significant bits (last bits) of some color channel of each pixel in the given image container. Since pixels are adjusted independently of each other, the LSB algorithm alters the distribution of the least significant bits, thereby simplifying detection of the payload. A modification of this method, which does not substantially alter the distribution of the least significant bits, is a so-called $\pm 1$-embedding (Ker, 2005). This approach randomly adds or subtracts 1 from some color channel pixel so that the last bits would match the ones needed. In this paper we basically consider the $\pm 1$-embedding algorithm.

There are more sophisticated algorithms for information embedding to raster images: WOW (Holub & Fridrich, 2012), HUGO (Pevny et al., 2010), S-UNIWARD (Holub et al., 2014), and others. They are derived from key ideas of the LSB algorithm, but utilize a more strategic pixel manipulation technique: for the raw image $X$ and its final version with a secret message $\hat{X}$ the pixels are picked in such a way as to minimize the distortion function

$$D(X, \hat{X}) = \sum_{i=1}^{n_1} \sum_{j=1}^{n_2} \rho(X_{ij}, \hat{X}_{ij}) |X_{ij} - \hat{X}_{ij}|,$$

where $\rho(X_{ij}, \hat{X}_{ij})$ is the cost of changing pixel of $X$, specific for each particular steganography algorithm.

For detecting presence of hidden information in the container Steganalysis is usually used. The stage which distinguishes images with some hidden message from empty is usually performed by binary classification. The basic approach to steganalysis is based on feature extractors (such as SPAM (Pevnỳ et al., 2010), SRM (Fridrich & Kodovskỳ, 2012), etc.) combined with traditional machine learning classifiers, such as SVM, decision trees, ensembles etc. With the recent overwhelming success of deep neural networks, newer neural network based approaches to steganalysis are gaining popularity, Qian et al. (2015b). For example, in Pibre et al. (2015) authors propose to use deep convolution neural networks (CNN) for steganalysis and show that classification accuracy can be significantly increased while using CNN instead of usual classifiers.

## 3 ADVERSARIAL NETWORKS

Generative Adversarial Networks (GAN) is a recent approach to deep unsupervised learning, proposed in 2014 in Goodfellow et al. (2014), which is capable of dynamically representing a sampler from input data distribution and generate new data samples.

The main idea of such approach to learning is that two neural networks are trained simultaneously:

- a generative model ($G$) that receives noise from the prior distribution $p_{noise}(z)$ on input and transforms it into a data sample from the distribution $p_g(x)$ that approximates $p_{data}(x)$;
- a discriminative model ($D$) which tries to detect if an object is real or generated by $G$.

The learning process can be described as a minimax game: the discriminator $D$ maximizes the expected log-likelihood of correctly distinguishing real samples from fake ones, while the generator $G$ maximizes the expected error of the discriminator by trying to synthesize better images. Therefore during the training GAN solve the following optimization problem:

$$L(D, G) = \mathbb{E}_{x \sim p_{data}(x)} \left[ \log D(x) \right] + \mathbb{E}_{z \sim p_{noise}(z)} \left[ \log(1 - D(G(z))) \right] \to \min_G \max_D, \quad (1)$$

where $D(x)$ represents the probability that $x$ is a real image rather then synthetic, and $G(z)$ is a synthetic image for input noise $z$.

Coupled optimization problem (1) is solved by alternating the maximization and minimization steps: on each iteration of the mini-batch stochastic gradient optimization we first make a gradient ascent step on $D$ and then a gradient descent step on $G$. If by $\theta_M$ we denote the parameters of the neural network $M$, then the update rules are:

- Keeping the $G$ fixed, update the model $D$ by $\theta_D \leftarrow \theta_D + \gamma_D \nabla_D L$ with

$$\nabla_D L = \frac{\partial}{\partial \theta_D} \left\{ \mathbb{E}_{x \sim p_{data}(x)} \left[ \log D(x, \theta_D) \right] + \mathbb{E}_{z \sim p_{noise}(z)} \left[ \log(1 - D(G(z, \theta_G), \theta_D)) \right] \right\},$$
$$(2)$$

- Keeping $D$ fixed, update $G$ by $\theta_G \leftarrow \theta_G - \gamma_G \nabla_G L$ where

$$\nabla_G L = \frac{\partial}{\partial \theta_G} \mathbb{E}_{z \sim p_{noise}(z)} \left[ \log(1 - D(G(z, \theta_G), \theta_D)) \right] . \tag{3}$$

In Radford et al. (2015) the GAN idea was extended to deep convolutional networks (DCGAN), which are specialized for image generation. The paper discusses the advantages of adversarial training in image recognition and generation, and give recommendations on constructing and training DCGANs. In fig. 1 we depict a sample of synthetic images of a freshly trained DCGAN on the Celebrities dataset (Ziwei Liu & Tang, 2015). The images indeed look realistic, albeit with occasional artifacts.

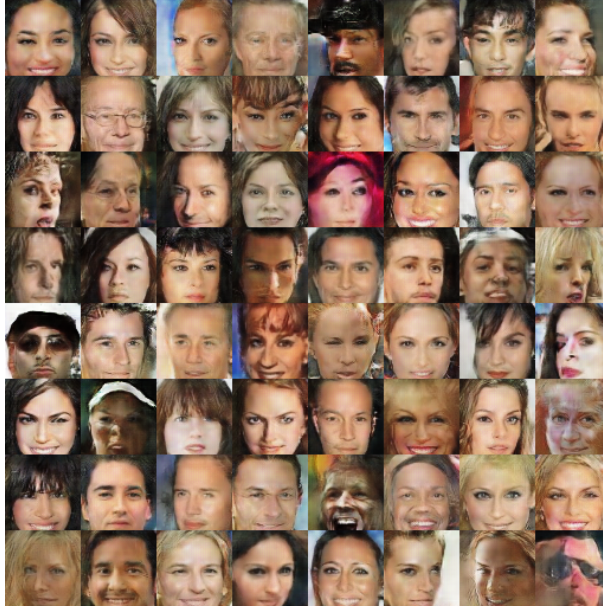

Figure 1: Sample synthetic images generated by DCGAN

# 4 STEGANOGRAPHIC GENERATIVE ADVERSARIAL NETWORKS

In order to apply GAN methodology to steganographic applications, we introduce Steganographic Generative Adversarial Networks model (SGAN), which consists of

- a generator network $G$, which produces realistic looking images from noise;
- a discriminator network $D$, which classifies whether an image is synthetic or real;
- a discriminator network $S$, the steganalyser, which determines if an image contains a concealed secret message.

By $Stego(x)$ we denote the result of embedding some hidden message in the container $x$.

Since we want the generator to produce realistic images that could serve as containers for secure message embedding, we force $G$ to compete against the models $D$ and $S$ simultaneously. If we denote by $S(x)$ the probability that $x$ has some hidden information, then we arrive at the following game:

$$
\begin{aligned}
L = \alpha \Big( &\mathbb{E}_{x \sim p_{data}(x)} \left[ \log D(x) \right] + \mathbb{E}_{z \sim p_{noise}(z)} \left[ \log(1 - D(G(z))) \right] \Big) + \\
&+ (1 - \alpha) \mathbb{E}_{z \sim p_{noise}(z)} \left[ \log S(Stego(G(z))) + \log(1 - S(G(z))) \right] \to \min_G \max_D \max_S .
\end{aligned} \tag{4}
$$

We use a convex combination of errors of $D$ and $S$ with parameter $\alpha \in [0, 1]$, which controls the trade-off between the importance of realism of generated images and their quality as containers

against the steganalysis. Analysis of preliminary experimental results showed that for $\alpha \leq 0.7$ the generated images are unrealistic and resemble noise.

The full scheme of SGAN is presented in fig. 2. Each arrows represent output- input data flows.

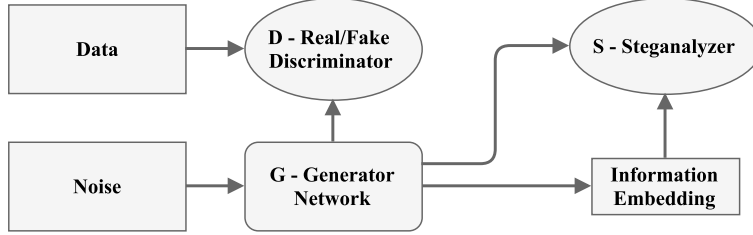

Figure 2: SGAN information flow diagram

Stochastic mini-batch Gradient descent update rules for components of SGAN are listed below:

- for $D$ the rule is $\theta_D \leftarrow \theta_D + \gamma_D \nabla_G L$ with

$$\nabla_G L = \frac{\partial}{\partial \theta_D} \Big\{ \mathbb{E}_{x \sim p_{data}(x)} \big[ \log D(x, \theta_D) \big] + \mathbb{E}_{z \sim p_{noise}(z)} \big[ \log(1 - D(G(z, \theta_G), \theta_D)) \big] \Big\} ;$$

- for $S$ (it is updated similarly to $D$): $\theta_S \leftarrow \theta_S + \gamma_S \nabla_S L$ where

$$\nabla_S L = \frac{\partial}{\partial \theta_S} \mathbb{E}_{z \sim p_{noise}(z)} \big[ \log S(Stego(G(z, \theta_G)), \theta_S) + \log(1 - S(G(z, \theta_G), \theta_S)) \big] ;$$

- for the generator $G$: $\theta_G \leftarrow \theta_G - \gamma_G \nabla_G L$ with $\nabla_G L$ given by

$$\nabla_G L = \frac{\partial}{\partial \theta_G} \alpha \mathbb{E}_{z \sim p_{noise}(z)} \big[ \log(1 - D(G(z, \theta_G), \theta_D)) \big]$$
$$+ \frac{\partial}{\partial \theta_G} (1 - \alpha) \mathbb{E}_{z \sim p_{noise}(z)} \big[ \log(S(Stego(G(z, \theta_G), \theta_S))) \big]$$
$$+ \frac{\partial}{\partial \theta_G} (1 - \alpha) \mathbb{E}_{z \sim p_{noise}(z)} \big[ \log(1 - S(G(z, \theta_G), \theta_S)) \big] .$$

The main distinction from the GAN model is that we update $G$ in order to maximize not only the error of $D$, but to maximize the error of the linear combination of the classifiers $D$ and $S$.

## 5 EXPERIMENTS

### 5.1 DATA DESCRIPTION

In our experiments [1] we use the Celebrities dataset (Ziwei Liu & Tang, 2015) that contains $200\,000$ images. All images were cropped to $64 \times 64$ pixels.

For steganalysis purposes we consider $10\%$ of data as a test set. We denote the train set by $A$, the test set by $B$ and steganography algorithms used for hiding information by $Stego(x)$. After embedding some secret information we get the train set $A + Stego(A)$ and the test set $B + Stego(B)$, We end up with $380\,000$ images for steganalysis training and $20\,000$ for testing. For training the SGAN model we used all $200\,000$ cropped images. After $8$ epochs of training our SGAN produces images displayed in fig. 3.

For information embedding we use the $\pm 1$-embedding algorithm with a payload size equal to $0.4$ bits per pixel for only one channel out of three. As a text for embedding we use randomly selected excerpts from some article from The New York Times.

---

[1]Code is available at https://github.com/dvolkhonskiy/adversarial-steganography

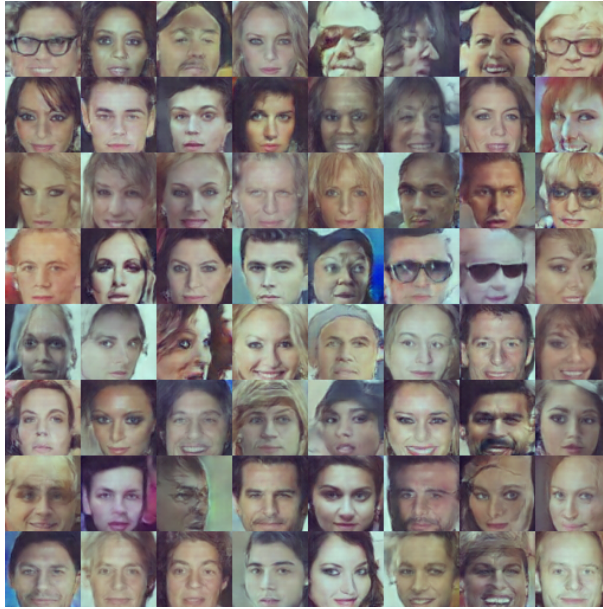

Figure 3: Examples of images, generated by SGAN after training for 8 epochs on the Celebrities dataset

## 5.2 EXPERIMENTAL SETUP

In this section we describe the SGAN model structure. By *C2D-BN-LR* we denote the following structural block of a convolutional neural network: Conv2d → Batch Normalization → Leaky ReLU.

The Steganalyser network $S$ and the Image Discriminator network have similar structure: four *C2D-BN-LR* layers, then a fully connected layer (1 neuron) → Sigmoid function is used to compute an output. The Image generator network $G$ is (in order) a fully-connected layer (8192 neurons), four *C2D-BN-LR* with Fractional-Strided convolution, then the Hyperbolic tangent function layer is used to compute normalised output.

The SGAN model is trained to solve (4) using the Adam optimization algorithm (Kingma & Ba, 2014) with the learning rate $2^{-4}$ and update parameters $\beta_1 = 0.5$ and $\beta_2 = 0.999$. For each mini-batch of images we update weights of $D$ and $S$ once, then we update weights of $G$ twice.

In the following experiments, in addition to the steganalyser $S$ we use an independent steganalyser $S^*$. We define a filter $F^{(0)}$ that is special for steganalysis applications (see Pibre et al. (2015), Qian et al. (2015a), Tan & Li (2014)), as follows

$$F^{(0)} = \frac{1}{12} \begin{pmatrix} -1 & 2 & -2 & 2 & -1 \\ 2 & -6 & 8 & -6 & 2 \\ -2 & 8 & -12 & 8 & -2 \\ 2 & -6 & 8 & -6 & 2 \\ -1 & 2 & -2 & 2 & -1 \end{pmatrix}.$$

The structure of the individual steganalyser $S^*$ has the form: 2D convolution with $F^{(0)}$ filter → Conv2D → Conv2D → Max Pooling → Conv2D → Conv2D → Max Pooling → Fully connected layer (1024 neurons) → Fully connected layer (1 neuron) → Sigmoid function for output. This structure provides state-of-the-art steganalysis accuracy, Pibre et al. (2015), and the filter $F^{(0)}$ allows to increase convergence speed of the steganalyser $S^*$ training.

For training of this steganalyser we use the Adam optimization algorithm on the loss (4) with the learning rate equal to $5^{-6}$, $\beta_1 = 0.9$, $\beta_2 = 0.999$. As a loss function we use a binary cross-entropy.

The setup of experiments can be described as follows:

- We train and use the SGAN and/or DCGAN model to generate images to be used as containers;

- We train the independent steganalyser $S^*$ using either real images (sec. 5.3) or generated images (sec. 5.4);

- We measure the accuracy of the steganalyser $S^*$.

## 5.3 TRAINING/TESTING ON REAL IMAGES

In this set of experiments we train the independent steganalyser $S^*$ on real images. Results are provided in tab. 1. From the results we conclude that even the usual DCGAN generate synthetic

Table 1: Accuracy of the steganalyser $S^*$ trained on real images

| Type of a test set \ Image generator | SGANs | DCGANs |
|---|---|---|
| Real images | 0.962 | |
| Generated images | 0.501 | 0.522 |

container images, that can easily deceive a steganalyser. Although containers generated by on the SGAN model, which is specially designed to deceive the steganalyser, improve on the result of DCGAN, the improvement is not significant. These results also suggest that $S^*$ may have been overfit to the data distribution of the real images, and failed to generalize to the GAN approximation of the data distribution in the synthetic image domain.

Another interesting result follows from an experiment with HUGO steganography algorithm and SGAN generated containers:

- We use the $\pm 1$-embedding algorithm for training the SGAN model;

- We test the HUGO steganography analyzer on real images and on images, generated by the SGAN model.

Containers generated by the SGAN model decrease HUGO steganalysis accuracy from 0.624 to 0.499, which demonstrates that the Steganographic Generative Adversarial Networks can potentially be used as a universal tool for generating Steganography containers tuned to deceive any specific steganalysis algorithm.

## 5.4 TRAINING/TESTING ON GENERATED IMAGES

In this set of experiments we train the steganalyser $S^*$ on images, generated using the DCGAN model from the prior noise distribution $p_{noise}(z)$ for some fixed seed value. In all experiments the size of the train set is fixed at $\approx 160\,000$. After training we test the analyser $S^*$ on images generated with the DCGAN model according to the following experimental setups:

**C1**. We use the same seed value;

**C2**. We use some randomly selected seed value;

**C3**. We use the same seed value, as in **C2**, and we additionally tune the DCGAN model for several epochs.

Table 2: Training/testing on generated images according to experimental conditions **C1**-**C3**

| Experimental Conditions | Accuracy |
|---|---|
| **C1** | 0.982 |
| **C2** | 0.517 |
| **C3** | 0.499 |

The experiment results in tab. 2 indicate that using different seed values when generating containers most likely affects the distribution of pixel value bits in such a way as make it easier to deceive

the steganalyser, fitted to another distribution in the train sample. Additional tuning of the image generator $G$ make this effect even more pronounced.

In the next set of experiments we train and test the steganalyser $S^*$ on images, generated according to the following experimental conditions:

**C4**. We generate a train set for the steganalyser $S^*$ using several different randomly selected seed values, and when generating the test set we use another fixed seed value;

**C5**. We generate the train set and the test set using a number of different randomly selected seed values;

**C6**. We use the same train and test sets, as in **C5**, and we additionally train the DCGAN model for several epochs.

Table 3: Training/testing on generated images according to experimental conditions **C4**-**C6**

| Experimental Conditions | Accuracy |
|---|---|
| **C4** | 0.649 |
| **C5** | 0.630 |
| **C6** | 0.581 |

According to tab. 3 the accuracy in case **C5** is lower than in the **C4** case, which can be explained by the test set of **C5** having more variability, being generated with different randomly selected seed values. Similarly, the accuracy in the **C4** case is higher than in **C2**, since in **C4** the train set was generated with several different randomly selected seed values, and thus is more representative. These observations confirm out initial conclusions, drawn from tab. 2.

We also conduct an experiment with classification of generated images without steganographic embeddings. For this purposes we train a DCGAN conditional model on the MNIST dataset, and train a separate classifier for the MNIST classification task. The trained classifier achieved almost perfect accuracy both on the held-out real MNIST dataset, and on synthetic images produced by the DCGAN. This provides evidence that it is possible to train an image classifier that shows acceptable accuracy both on real and synthetic images. However it is the artificial generation of image containers that breaks the usual approaches to steganalysis.

## 6 CONCLUSIONS AND FUTURE WORK

In this work

1. We open a new field for applications of Generative Adversarial Networks, namely, container generation for steganography applications;

2. We consider the $\pm1$-embedding algorithm and test novel approaches to more steganalysis-secure information embedding:
   a) we demonstrate that both SGAN and DCGAN models are capable of decreasing the detection accuracy of a steganalysis method almost to that of a random classifier;
   b) if we initialize a generator of containers with different random seed values, we can even further decrease the steganography detection accuracy.

In future, We plan to test our approach on more advanced steganographic algorithms, e.g. WOW (Holub & Fridrich, 2012), HUGO (Pevny et al., 2010) and S-UNIWARD (Holub et al., 2014).

ACKNOWLEDGMENTS

The research was supported solely by the Russian Science Foundation grant (project 14-50-00150). The authors would like to thank I. Nazarov for his assistance in preparation of this paper.

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
