# Peer review of "Generative Adversarial Networks for Image Steganography"

_ICLR 2017 — rejected_

[Official Review · AnonReviewer3 · rating 4 · confidence 3 · 13 Dec 2016]
**Official review for 'Generative Adversarial Networks for Image Steganography'**

I reviewed the manuscript as of December 6th.

Summary:
The authors build upon generative adversarial networks for the purpose of steganalysis -- i.e. detecting hidden messages in a payload. The authors describe a new model architecture in which a new element, a 'steganalyser' is added a training objective to the GAN model.

Major Comments:
The authors introduce an interesting new direction for applying generative networks. That said, I think the premise of the paper could stand some additional exposition. How exactly would a SGAN method be employed? This is not clear from the paper. Why does the model require a generative model? Steganalysis by itself seems like a classification problem (i.e. a binary decision if there a hidden message?) Would you envision that a user has a message to send and does not care about the image (container) that it is being sent with? Or does the user have an image and the network generates a synthetic version of the image as a container and then hide the message in the container? Or is the SGAN somehow trained as a method for detecting hidden codes performed by any algorithm in an image? Explicitly describing the use-case would help with interpreting the results in the paper.

Additionally, the experiments and analysis in this paper is quite light as the authors only report a few steganalysis performance numbers in the tables (Table 1,2,3). A more extensive analysis seems warranted to explore the parameter space and provide a quantitative comparison with other methods discussed (e.g. HUGO, WOW, LSB, etc.) When is it appropriate to use this method over the others? Why does the seed effect the quality of results? Does a fixed seed correspond realistic scenario for employing this method?

Minor comments:
- Is Figure 1 necessary?
- Why does the seed value effect the quality of the predictive performance of the model?

[Official Review · AnonReviewer2 · rating 6 · confidence 4 · 16 Dec 2016]
**Creative and promising idea; presentation should be improved**

I found this paper very original and thought-provoking, but also a bit difficult to understand. It is very exciting to see a practical use case for image-generating GANs, with potentially meaningful benchmarks aside from subjective realism.

I found eq. 4 interesting because it introduces a potentially non-differentiable black-box function Stego(...) into the training of (S, G). Do you in fact backprop through the Stego function?

- For the train/test split, why is the SGAN trained on all 200k images? Would it not be cleaner to use the same splits for training SGAN as for "steganalysis purposes"? Could this account for the sensitivity to random seed shown in table 2?
- Sec. 5.3: "Steganographic Generative Adversarial Networks can potentially be used as a universal tool for generating Steganography containers tuned to deceive any specific steganalysis algorithm.". This experiment showed that SGAN can fool HUGO, but I do not see how it was "tuned" to deceive HUGO, or how it could be tuned in general for a particular steganalyzer.

Although S* seems to be fooled by the proposed method, in general for image generation the discriminator D is almost never fooled. I.e. contemporary GANs never converge to actually fooling the discriminator, even if they produce samples that sometimes fool humans. What if I created an additional steganalyzer S**(x) = S*(x) * D(x)? This I think would be extremely difficult to fool reliably because it requires realistic image generation.

After reading the paper several times, it is still a bit unclear to me how or why precisely one would use a trained SGAN. I think the paper could be greatly improved by detailing, step by step, the workflow of how a hypothetical user would use a trained SGAN. This description should be aimed at a reader who knows nothing or very little about steganography (e.g. most of ICLR attendees).

[Official Review · AnonReviewer1 · rating 5 · confidence 3 · 20 Dec 2016]

This paper proposes an interesting application of the GAN framework in steganography domain. In addition to the normal GAN discriminator, there is a steganalyser discriminator that receives the negative examples from the generator and positive examples from the generator images that contain a hidden payload. As a result, the generator, not only learn to generate realistic images by fooling the discriminator of the GAN, but also learn to be a secure container by fooling steganalyser discriminator. The method is tested by training an independent steganalyser S* on real images and generated images.

Given that in the ICLR community, not many people are familiar with the literature of steganography, I think this paper should have provided more context about how exactly this method can be used in practice, what are the related works on setganalysis-secure message embedding and probably a more thorough sets of experiments on more than one dataset.

The proposed SGAN framework (Figure 2) does make sense to me, and I think it is very general and can have more applications other than the steganography domain. But it is not clear to me why fooling the steganalyser discriminator S, necessarily mean that we can fool an independent discriminator S*?

Also I find it surprising that a different seed value, can make such a huge difference in the accuracy.

In short, the ideas of this paper are interesting and potentially useful, but I think the presentation of this paper should be improved so that it becomes more suitable for the ICLR and machine learning community.

[Final Decision · Program Chairs · 06 Feb 2017]
**ICLR committee final decision**

This paper examines an application of that deviates from the usual applications presented at ICLR. The idea seems very interesting to the reviewers, but a number of reviewers had trouble really understanding why the proposed SGAN would be attractive for this problem, and this problem setup with the SGAN in general. Clearer concrete 'use case scenarios' and experimentation that helps clarify the precise application setting and the advantages of the SGAN formulation would help make this work more impactful on the community. Given the quality of other paper submitted to ICLR this year the reviewer scores are just short of the threshold for acceptance